# Hepatoblastoma Relapse—Findings from the German HB99 Trial and the German Liver Tumor Registry

**DOI:** 10.3390/cancers16040696

**Published:** 2024-02-06

**Authors:** Rebecca Maxwell, Beate Häberle, Roland Kappler, Dietrich von Schweinitz, Mark Rassner, Julia von Frowein, Irene Schmid

**Affiliations:** 1Department of Pediatrics, Dr. von Hauner Children’s Hospital, University Hospital, LMU Munich, Lindwurmstraße 4, 80337 Munich, Germany; rebecca.maxwell@med.uni-muenchen.de (R.M.); mark.rassner@med.uni-muenchen.de (M.R.); julia.frowein@med.uni-muenchen.de (J.v.F.); 2Department of Pediatric Surgery, Dr. von Hauner Children’s Hospital, University Hospital, LMU Munich, 80337 Munich, Germany; beate.haeberle@med.uni-muenchen.de (B.H.); roland.kappler@med.uni-muenchen.de (R.K.); dietrich.schweinitz@med.uni-muenchen.de (D.v.S.)

**Keywords:** hepatoblastoma, relapse, AFP, irinotecan

## Abstract

**Simple Summary:**

The outcome for patients who relapse following initial successful treatment of hepatoblastoma (HB) remains poor according to the limited amount of available data, much of which is greater than 10 years old. There are no guidelines for the treatment of these patients. The aim of this work was to increase our knowledge about relapse patients and their current treatment to gain insights that can contribute to treatment guidelines and, thus, improvements in outcomes for these patients in the future. Our review of records from 362 HB patients treated between 1999 and 2018 identified 25 relapse patients. Detailed analysis of their data revealed a number of interesting findings, including factors such as the occurrence of late relapses, low AFP levels at relapse, and possible methods for improvements in treatment strategies. A total of 64% of our HB-relapse patients went on to achieve a second complete remission. The 3-year event-free survival was 63%.

**Abstract:**

Survival rates for HB patients have improved; however, outcomes for patients who relapse remain poor. A retrospective review of information gathered for the HB99 study and the German Liver Tumor Registry identified 25 relapse patients (6.9%, 25/362). The median time from initial diagnosis to first relapse was 13 months (range: 5–66 months). Two patients relapsed >36 months after initial diagnosis. A total of 68% (17/25) of relapses were metastatic, 24% local, and 8% combined. 67% of local relapses were alive at the last follow-up, in contrast to 53% of metastatic and 0% of combined relapses. At the last follow-up, 73% (8/11) of patients with lung relapses were still alive (0/4 with peritoneal, 1/2 with CNS involvement). A total of 20% of the patients had AFP-negative relapses, 64% of the relapse patients achieved a second complete remission, 69% were still in complete second remission at the last follow-up (median FU of 66 months), and 83% (5/6) of irinotecan-naïve patients who received relapse treatment including irinotecan were in second complete remission at the last follow-up. The 3-year overall survival/event-free survival from relapse was 63%/48% respectively. There is a good chance that HB patients will achieve a second remission despite a first relapse. However, patients who suffer further relapses tend to have a poorer prognosis.

## 1. Introduction

Hepatoblastoma (HB), although rare in absolute terms (1.2–1.5 people per million per year [1]), is the most common primary malignant liver tumor in infants and small children [2]. Study-driven refinements in the treatment of HB over time have led to significant improvements in outcomes from about 30% in the early 1980s to >80% now [3,4,5]. Despite these advances, however, the outcomes for those patients who relapse after an initial remission, according to currently available data, remain poor (SIOPEL 1–3 trials: 3-year event-free survival (EFS) of 34% and 3-year overall survival (OS) of 43% [6]). Due to the rarity of this condition, there are no general guidelines as to how to treat relapse HB patients. In addition, there are only a small number of articles providing insights into this area. The majority of these are case reports or single-center studies (e.g., [7]). The three most recent, larger studies of note in this area are based on results from the International Childhood Liver Tumor Strategy Group (SIOPEL) ([6]: 59 relapse patients) as noted above, and from the Children’s Oncology Group (COG) ([8]: 71 patients, relapse + progression; [9]: 30 patients, relapse + progression). The SIOPEL study is based on data collected between 1990 and 2004, and the two COG studies, whilst slightly more recent (data from COG publications between 2000 and 2019), include both relapse and progression patients.

In comparison, our data include relapse patients only, recruited between 1999 and 2018 and principally treated according to either the HB99 study or the SIOPEL 3/4 studies. Thus, although we have a relatively small sample size, which unfortunately precludes extensive statistical analysis, we can provide information to enhance the current amount of available data and validate those findings from the SIOPEL trial. We can also use the data collected to make observations with regard to which of the treatment alterations over this time may have led to our observed improvement in survival outcomes.

## 2. Materials and Methods

Our review included patients from the HB99 study [5] and the Liver Tumor Registry (LTR) who had had an HB tumor relapse, as evidenced by imaging, after complete remission (CR). Both the LTR and the HB99 study prospectively recruited/still recruit patients from the German Society for Pediatric Oncology and Hematology (GPOH) centers in Germany, Austria, and Switzerland. The methods used in the HB99 study (1 January 1999–31 December 2008) have already been published [5]. The LTR started recruiting new cases of pediatric liver tumors (patients < 18 years) from GPOH centers on 17 January 2011 and is still ongoing. Information is collected at the treatment center and then sent to Ludwig-Maximilians-University (LMU) in Munich, where it is entered into a database. Clinical data were extracted retrospectively from the aforementioned datasets. Informed consent for data collection was obtained at the time of recruitment. Ethical approval for the HB99 study was granted by the Children’s University Hospital in Basal and the Rheinische-Friedrich-Wilhelms-University in Bonn. Ethical approval for the LTR was granted by the LMU in Munich.

The treatment that the patients received at primary diagnosis varied according to the standard treatments used at the time of occurrence of this initial treatment. For this study, we stratified patients as either standard-risk (SR) or high-risk (HR) according to the pre-treatment extent of disease (PRETEXT) system [5,10]. SR patients were those who had tumors confined to ≤3 segments of the liver and no additional risk factors (i.e., AFP > 100 ng/mL, no distant metastases (M-), no extrahepatic intra-abdominal tumors (E-), no vessel involvement (V-, P-), unifocal tumors (F-), no tumor rupture (R-), and no affected lymph nodes (N-)). All other patients were classified as HR. The standard chemotherapy treatment in the HB99 trial was split into two risk-based groups [5]. SR patients received ifosfamide, cisplatin, and doxorubicin (IPA). HR patients initially received carboplatin and etoposide (CE) at the conventional dose, with responders then going on to receive high-dose (HD) CE followed by an autologous stem-cell transplantation, whilst non-responders went on to receive IPA instead. The standard chemotherapy given to the majority of patients whose data were collected in the LTR was a SIOPEL 3- or 4-based protocol, again with two risk-stratified treatment groups. The SR patients received cisplatin monotherapy [11], and the HR patients received cisplatin, carboplatin, and doxorubicin [12,13]. There was no standardized approach to the treatment of relapsed patients during this time period.

Tumors were usually resected following one of the planned chemotherapy cycles or after all of the chemotherapy cycles from the chosen protocol had been given. Lung metastases (mets) were resected if still reliably measurable on the CT scan after neoadjuvant chemotherapy. Unresectable liver tumors were treated via liver transplantation after surgical excision of radiologically identified palpable lung mets where applicable. The extent of tumor resection was classified according to available data (surgery +/− histology reports). R0 resections are those where there was a microscopically complete resection, R1 resections were microscopically incomplete resections, and R2 resections were macroscopically incomplete resections. A rescue liver transplant is one performed after a previous resection/transplant, whereas a primary liver transplant is the first liver operation performed.

Routine control examinations during follow-up, after the achievement of complete remission, typically included AFP measurements, abdominal ultrasound scans, and chest X-rays. The frequency of these examinations reduced over time (e.g., 1st–2nd year: 3 monthly, 3rd year: 6 monthly, and 4th–9th year: yearly).

Relapse was defined as the appearance of a new tumor lesion on imaging, with or without an increase in AFP, after the previous achievement of complete remission. An AFP rise without radiological findings was not counted as a relapse. Complete remission was defined as the completion of treatment with the disappearance of all identified tumor lesions combined with at least one normal AFP (for age) result. Overall survival (OS) is the time between the date of a specific given time point (i.e., diagnosis or relapse) and the date of either death from any cause or the date of last follow-up (FU). Event-free survival (EFS) is the time from the date of a given event (i.e., diagnosis or relapse) to the date of the next event post-relapse (i.e., a second relapse, the start of progression, death, or the date of last FU). Progression post-first-relapse was defined as the growth of the tumor and/or an increase in AFP in those patients who had not yet achieved a second complete remission.

Statistical analysis was performed using SPSS Version 28. Kaplan–Meier survival plots were used to calculate OS and EFS figures [14]. The use of statistical analysis in this paper was limited owing to the small number of patients included in this study.

## 3. Results

### 3.1. Relapse Rate and Initial Characteristics and Treatment of Relapse Patients

The relapse rate amongst our patients was 6.9% (HB99 study: 12/142 (8.5%), LTR 13/220 (5.9%)). The characteristics of the 25 relapse patients at diagnosis have been summarized in Table 1.

All the patients who went on to relapse were initially treated both surgically and with chemotherapy (see Table 2).

Of the eight F+ patients at diagnosis, two out of eight had a primary liver transplant (subsequent local relapse: 50%), and six out of eight had initial resections (subsequent local relapse: 50%).

Of the 10 patients who had lung mets at diagnosis and the 3 who developed the mets during treatment, 9/13 had lung met resections prior to remission (subsequent lung relapse: 67%), whilst 4/13 were treated with chemotherapy alone (subsequent relapse: 50%).

The majority of patients had ≥21 days (17/25 (68%)) between their initial liver operation and their next chemotherapy treatment. A total of 10/17 (59%) of those who had a ≥21 days are dead as compared to 2/8 (25%) who had <21 days. This difference was not statistically significant (*p* = 0.207, log-rank (Mantel–Cox)).

The median time from initial diagnosis to the initial liver resection was 3.5 months (range: 0–6 months). A total of 3/25 patients were operated on at <2 months post diagnosis (OS: 0%), 6/25 at 2–<3 months post diagnosis (OS: 50%), 5/25 at 3–<4 months post diagnosis (OS: 100%), 8/25 at 4–<5 months post initial diagnosis (OS: 50%), 2/25 at 5–<6 months post diagnosis (OS: 50%), and 1/25 at ≥6 months post-diagnosis (OS: 0%). The groups were too small for further statistical analysis.

### 3.2. Details of Relapse

The median time from initial diagnosis to first relapse was 13 months (range: 5–66 months). Two patients relapsed >36 months after initial diagnosis (8%) (see Table 2).

Metastatic relapses (68%) were more common than local (24%) or combined (8%) relapses (see Table 3). A total of 4/6 (67%) local relapse patients were still alive at the last FU as compared to 9/17 (53%) with metastatic relapses and 0/2 with combined relapses. Of the 17 patients with mets, 65% (11/17) had lung mets, 24% (4/17) had peritoneal mets, and 12% (2/17) had cerebral mets, with 73% (8/11) vs. 0% vs. 50% still alive at the last FU, respectively. There was a statistically significant difference in overall survival from relapse between these three groups (*p* = 0.008, log-rank (Mantel–Cox)).

A total of 5/25 patients had an AFP ≤ 10 ng/mL at relapse, 5/25 patients had an AFP between 11 and 100 ng/mL, and 14/25 had an AFP > 101 ng/mL (1 patient: missing units). Patients who had an AFP of ≤10 ng/mL at the time of relapse had a significantly lower survival rate than those with a higher AFP (*p* = 0.033, log-rank (Mantel–Cox)). The median AFP level at relapse was 174 ng/mL (5–119,674 ng/mL).

The majority (20/25) of the relapse patients were once again treated with both chemotherapy and surgical resection (see Table 3). The five patients who did not receive both treatment options have all subsequently died. A total of 9/20 patients had surgery before chemo, and 11/20 after chemo had started. Notably, 5/9 (56%) of the surgery-first patients were still alive at the last FU versus 8/11 (73%) of the chemo-first patients; however, this was not a statistically significant difference (*p* = 0.344, log-rank (Mantel–Cox)).

A multitude of different chemotherapy regimens were used to treat the 22 relapse patients who received chemotherapy (see Table 3). A total of 18/22 patients were given at least one new agent in comparison to the initial treatment. From the five out of seven doxorubicin-naïve patients who received it in combination with other agents, three out of five (60%) were still in second CR (complete remission) at the last FU. Of the six out of seven carboplatin-naïve patients were given it in combination with other agents, three out of six (50%) still in second CR at the last FU. Similarly, 4/15 ifosfamide-naïve patients were given it in combination with other agents, with 2/4 (50%) still in second CR at the last FU. Incomparison, 6/19 irinotecan-naïve patients were given it in combination with other agents, with 5/6 (83%) still in second CR at the last FU. Those patients who did not receive irinotecan during follow-up appeared to have a poorer OS from relapse than those who did (see Figure 1A). This difference was not statistically significant (*p* = 0.118, log-rank (Mantel–Cox)).

A total of 10/22 patients had ≤21 days and 8/22 patients had >21 days between their first op. post-relapse and their next chemo. Additionally, 4/22 patients had an op. that was not followed by chemotherapy (two transplants and two progressions). A total of 8/10 (80%) patients whose chemo started ≤21 days after their relapse op. were still alive at the last FU as compared to 4/8 (50%) patients whose chemo started >21 days after their relapse op. This difference was not statistically significant (*p* = 0.234, log-rank (Mantel–Cox)).

A total of 15/22 relapse patients who had surgery had initial relapse operations that were classed as R0, 4 as R1, and 2 as R2 (MD: 1 patient). In total, 12/15 (80%) of these R0 patients were still alive at the last FU as compared to 1/4 R1 and 0/2 R2 patients. This difference was statistically significant (*p* = 0.001, log-rank (Mantel–Cox)).

Three of the total eight transplants occurred during the first relapse, with two patients having their first transplant and one having their second. Two patients had their first transplant during their second relapse. Overall, only two out of five of these patients were alive at the last FU, both of whom had their first transplants after the first relapse. Two out of three (67%) of those F+ patients who had a transplant(s) were still alive at the last follow-up (second CR: one out of three (33%)) as compared to four out of five (80%) of those who did not have a transplant (second CR: 3/4 (75%)).

A total of 14/24 (58%) patients had an AFP drop after their initial relapse therapy (first 1–2 cycles of chemo or initial operation), 6/24 (25%) patients had an AFP increase, and 2/24 (8%) already had an AFP < 10 ng/mL pre-therapy (2/24 = MD). A total of 10/14 (71%) of the AFP drop patients were still alive at their last FU versus 2/6 (33%) of the AFP increase patients and 50% of the AFP < 10 ng/mL patients.

### 3.3. Outcome after Relapse

A total of 16/25 (64%) patients achieved a second complete remission (13 still alive at the last FU). Overall, 11/16 (69%) of these patients were still in a second CR at the last FU (median FU from the date of first relapse: 66 months, range: 15–186 months). A total of 5/16 patients (31%) had a further relapse (2/5 alive at the last FU). The median time from first relapse to second relapse was 14 months (range: 13–80 months). The median survival time from relapse until death for the 12 patients who died was 10 months (0–146 months). Seven died of disease progression after the first relapse, two died due to complications (CNS met, post-liver transplant), and three died following subsequent relapses.

The 3-year OS from relapse for all 25 relapse patients was 63% (95% confidence interval (CI): 41–79%), and the 3-year EFS was 48% (95% CI: 27–65%). The 5-year OS from relapse for all 25 relapse patients was 53%, and the 5-year EFS was 48% (see Figure 1B). There was no significant difference between the EFS and the OS from relapse in our group of 25 patients (see Figure 1B).

The one PRETEXT I patient was still alive at the last FU (second CR: 100%) as compared to 3/5 (60%) PRETEXT II patients (second CR: 100%), 4/10 (40%) PRETEXT III patients (second CR: 100%), and 5/9 (56%) PRETEXT IV patients (second CR: 60%) (see Figure 1C,D). Due to the small number of patients available in this study, we were unable to demonstrate a significant difference in the OS or EFS of patients from relapse within the study period based on their PRETEXT group assigned at diagnosis (EFS: *p* = 0.857, log-rank (Mantel–Cox)).

A total of 3/5 (60%) SR patients were still alive at the last FU (second CR: 100%) as compared to 10/20 (50%) of the HR patients (second CR: 8/10 (80%)). There was no significant difference in terms of OS from relapse between these groups of patients (*p* = 0.919, log-rank (Mantel–Cox)).

## 4. Discussion

This retrospective analysis indicated that there is a good chance that HB patients will achieve a second remission as our patients achieved a 3-year overall survival and event-free survival of 63% and 48%, respectively.

As noted in previous studies [6], patients who relapsed tended to have a slightly higher median age at the time of initial diagnosis, were more likely to be male, have a higher prognostic risk classification (PRETEXT IV, HR) and more additional risk factors (see Table 1) compared to the results for all HB patients at diagnosis.

Points of note from the initial treatment included firstly that more conservative surgical treatment options can still lead to good outcomes, whilst more aggressive treatment does not always improve outcomes. Thus, not all F+ patients needed to have a transplant to prevent a local relapse, and a transplant did not always prevent local relapse. This conclusion is similar to that of Fahy et al. 2019 [15]. In addition, our data suggest that not all patients with lung mets need to have a metastasectomy to prevent further lung relapses and that having a lung metastasectomy does not always prevent lung relapses. These results are similar to those of Zsiros et al. 2013 [13].

Secondly, it was noted that the timing of the surgery and chemotherapy during the initial treatment may influence survival rates. A smaller percentage (25% versus 59%) of our patients who had <21 days between their operation and their next chemotherapy subsequently died as compared to those who had ≥21 days. This difference was not statistically significant. The number of relapse patients who had a delay of ≥21 days (83% HB99 vs. 54% LTR) was reduced amongst the LTR patients as compared to the HB99 patients. The reduction in delay seen between the two studies resulted from changes in practice following the publication of articles that suggested that a shorter delay may lead to improved survival [16,17].

The results relating to the incidence and timing of relapse were also in accordance with previous studies [6]. Thus, we found a low relapse rate for HB of 6.9% (SIOPEL 1–3: 8.4% [6]) and a median time from the initial diagnosis to the first relapse of 13 months (SIOPEL 1–3: 12 months [6]) with the possibility of late relapses (8% ≥ 36 months vs. 10% SIOPEL 1–3 [6]).

Of interest here was firstly the reduction in the overall relapse rate between our two studies (8.5% HB99 versus 5.9% LTR) and their comparison with the SIOPEL 1–3 study [6] (1990–2004) with a relapse rate of 8.4%, which is similar to the more contemporaneous HB99 study (1999–2008).

Secondly, it was noted that at least 20% of our patients had an AFP ≤ 10 ng/mL at relapse (SIOPEL 1–3: 15%) and that a further 20% of patients had AFPs from 11 to 100 ng/mL. This highlights the potential risk of using regular AFP measurements alone as an alternative surveillance technique for the detection of recurrence [18] and indicates that even small rises in AFP may be important and should perhaps promote more rapid subsequent assessment of AFP levels. New studies assessing the AFP-L3 fraction may provide a potential means of more accurately identifying those patients with slightly raised AFPs who will relapse from those who will not [19].

The relapse characteristics and their prognostic value and the importance of certain elements of the relapse treatment were likewise as expected (e.g., [6]). The majority of relapses were metastatic (68%) and occurred in the lung (65%). Risk factors for poorer prognosis upon relapse included the presence of non-pulmonary metastatic relapses (*p* = 0.008) and AFPs of <10 ng/mL at diagnosis of relapse (*p* = 0.033). Those patients treated with a combination of surgery and chemotherapy seemed to do better than those who were not (13/ 20 alive at the end of FU vs. 0/5). An initial R0 relapse operation led to a significantly better (*p* = 0.001) overall survival than a ≥R1 relapse operation. A total of 77% of patients received either cisplatin or carboplatin, and 82% of patients were given at least one new agent in comparison to their initial treatment. This approach is in line with findings from the review of the available literature by Venkatramani et al. in 2012 [20]. An AFP drop after initial relapse therapy appeared to be associated with a better prognosis than an increase in AFP. Thus, changes in AFP levels following the start of relapse treatment appear to provide a prognostic guide for future outcomes, as has been found to be the case during the initial treatment of hepatoblastoma patients [21,22].

We feel that the following chemotherapy-related factors, whilst not statistically significant, are worthy of further investigation. Firstly, the finding that 73% (8/11) of patients with chemotherapy first, as compared to 56% (5/9) of patients with the operation first, were still alive at their last follow-up appears to suggest that chemotherapy first may lead to better, or at least equal, outcomes. Secondly, there may be a potential role for irinotecan in relapse compared to the other agents used in this study (see Figure 1A). There have been a number of articles that have suggested a benefit of irinotecan in relapse patients, and this study would seem to be in line with that view [6,23,24,25].

The findings relating to the outcomes of the transplant patients were of interest as well. Despite previous suggestions that patients who have rescue transplants tend to have a poorer prognosis [26], the two patients who had their first liver transplant after the first relapse were the only transplant patients in this study who were still in ongoing second CR at the last follow-up. Thus, the use of more conservative therapy during the initial treatment, which could then be supplemented with a rescue transplant at a later time point if needed, should not be automatically excluded upfront in favor of a primary liver transplant. This is especially the case given the longer-term implications of liver transplants [27].

A total of 64% of our patients achieved a second complete remission (SIOPEL 1–3: 52% [6]), with 69% still in second complete remission at the last FU (SIOPEL 1–3: 58% [6]). These results appear slightly better than those seen in the SIOPEL 1–3 trials. This apparent improvement may be a result of refinements in diagnosis and treatment over time. The advances in diagnosis (e.g., radiology and histology) include, in particular, the differentiation of small-cell undifferentiated disease (SCUD) [28] and hepatocellular neoplasm–not otherwise specified (HCN-NOS) patients [29] from the HB patients. The changes in the chemotherapy treatment include alterations in the management of chemoresistance, such as the identification and use of more effective alternative agents in relapse (e.g., doxorubicin and irinotecan). The improvements in the surgical treatment of HB include the reduction in post-op treatment delays and the ever-more skilled use of liver transplants. The modification to the overall approach includes the increasing use of both chemotherapy and surgery for the majority of patients.

A total of 52% of the relapse patients were alive at their last follow-up. The 3-year OS and EFS from relapse were 63% (95% CI: 41–79%) and 48% (95% CI: 27–65%), respectively (SIOPEL 1–3 [6]: 43% and 34%). There was no significant difference between OS and EFS, thus suggesting that those who go on to have future events are likely to have a poor prognosis (Figure 1B). Patient numbers were insufficient to assess whether initial risk stratifications (e.g., PRETEXT) were linked with patient outcomes.

## 5. Conclusions

This study provided further confirmation of previously published findings with regard to relapse in hepatoblastoma patients and suggested an improvement in overall survival rates as a result of the translation of research study findings since the SIOPEL 1–3 trials into clinical practice. The 3y-OS from relapse for this group of 25 relapse patients was 63%, and the 3y-EFS was 48%.

Findings of particular note included the fact that AFP levels alone are insufficient for detection of relapse in a significant minority of relapse patients, that the role of irinotecan in the treatment of hepatoblastoma relapse patients needs further investigation, that the relative timing of combined surgical and chemotherapeutic interventions appears to be important in relapse treatment, and finally that more conservative treatment options and rescue liver transplants should be considered as viable therapy strategies. These findings should be consolidated with other available data on relapse patients so as to help provide a future guideline for the treatment of these patients.

There is a good chance that HB patients will achieve a second remission despite a first relapse. However, patients who suffer further relapses are likely to have a poorer prognosis.

## Figures and Tables

**Figure 1 cancers-16-00696-f001:**
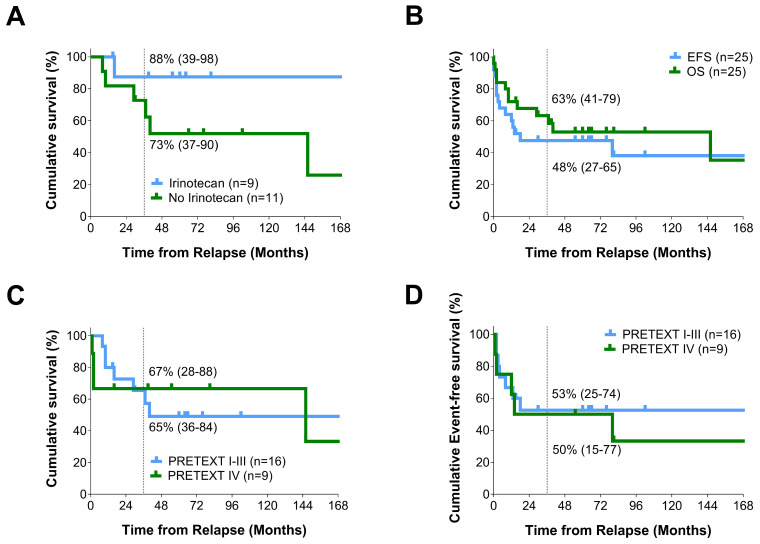
Survival Curves of Note for our Relapse Patients. (**A**) Overall-survival from relapse for those patients who were treated with irinotecan during relapse and those who were not. Those patients who did not receive irinotecan during follow-up appeared to have a poorer OS from relapse than those who did. The 2 patients who received palliative chemo have been removed from this analysis. This difference was not statistically significant (Log Rank (Mantel–Cox), *p* = 0.118); (**B**) Overall survival (OS) versus event free survival (EFS) from relapse for the HB relapse patients in this study. There was no significant difference between OS and EFS from relapse in this group of patients (Log Rank (Mantel–Cox), *p* = 0.447); (**C**) Overall survival (OS) from relapse for PRETEXT group I–III patients as compared to PRETEXT group IV patients. There was no significant difference in terms of OS from diagnosis between these groups of patients (Log Rank (Mantel–Cox), *p* = 0.797); (**D**) Event Free Survival (EFS) from relapse for PRETEXT group I-III patients as compared to PRETEXT group IV patients. There was no significant difference in EFS between these groups of patients (Log Rank (Mantel–Cox), *p* = 0.857).

**Table 1 cancers-16-00696-t001:** Overview of Clinical Characteristics of the 25 relapse patients. Data from the HB 99 trial and the SIOPEL 1–3 relapse paper [6], as well as provisional information from the LTR, have been provided for comparison.

	All Relapses (25) (6L, 17M, 2C)	SIOPEL 1–3 Relapses [6] (59) (21L, 32M, 5C, 1MD)	All HB99 Patients (142)	All LTR Patients (Till 30 August 2019) (220)	All SIOPEL 1–3 Patients [6] (695)
**Median Age and Range (months)**	24 (3–158)	MD 84L, 21.5M, 9C	16	18 (0–145)	17.2 (0–185)
**Gender (Male/Female)**	19/6 (3.2/1)	42/17 (2.5/1)	98/44 (2.2/1)	127/93 (1.4/1)	423/272 (1.6/1)
**PRETEXT**				MD 2	MD 7
**I**	1 (4%)	2 (3%)	4 (3%)	20 (9%)	34 (5%)
**II**	5 (20%)	15 (26%)	48 (34%)	73 (33%)	256 (37%)
**III**	10 (40%)	24 (41%)	70 (49%)	86 (39%)	266 (38%)
**IV**	9 (36%)	17 (29%)	20 (14%)	39 (18%)	132 (20%)
**Risk Group**				MD 2	MD 12
**SR**	5 (20%)	26 (45%)	85 (60%)	100 (45%)	430 (62%)
**HR**	20 (80%)	32 (55%)	57 (40%)	118 (54%)	253 (36%)
**Annotations Factors**					
**M+**	10 (40%)	17 (29%)	29 (21%) (MD 1)	40 (18%) (MD 11)	125 (18%)
**V+**	4 (16%)	4 (7%)	9 (6%)	23 (10%) (MD 21)	NA
**P+**	6 (24%)	7 (12%)	12 (9%)	28 (13%) (MD 13)	NA
**E+**	2 (8%)	NA	5 (4%) (MD 7)	NA	NA
**R+**	3 (12%)	NA	8 (7%) (MD 20)	NA	NA
**F+**	8 (32%)	21 (36%)	35 (25%) (MD 1)	71 (32%) (MD 2)	114 (16%) (MD 37)
**AFP at Diagnosis (ng/mL)**					
**≤100**	2 (8%)	NA	10 (7%)	NA	NA
**101 > 1 Mill**	23 (92%)	NA	132 (93%)	NA	NA
**Histology**				NA	NA
**Epithelial**	9 (36%)	28 (47%)	71 (50%)		
**Mixed**	7 (28%)	12 (20%)	48 (34%)		
**Macrotrabecular**	1 (4%)	3 (5%)	-		
**HB (DD HCC)**	4 (16%)	-	-		
**SCUD**	2 ^a^ (8%)	10 (17%)	4 (3%)		
**MD**	2 (8%)	-	-		

^a^ INI1: 1*positive, 1*MD. NA = not available, MD = missing data, L = local, M/M+ = metastatic, C = combined, PRETEXT = pre-treatment extent of disease, V+ = venous invasion, P+ = portal vein invasion, E+ = extrahepatic intra-abdominal mets, R+ = tumor rupture, F+ = multi-focal, HB = hepatoblastoma, HCC = hepatocellular carcinoma, SCUD = small-cell undifferentiated disease, DD = differential diagnosis.

**Table 2 cancers-16-00696-t002:** Age, PRETEXT Classification, Histology, Treatment, and Time to Relapse for the 25 Relapse Patients.

Pat. No.	Age (m)	PRETEXT	Histology	Chemo	Surgery	Time Dx to Relapse (m)
1 (D)	104	III N+	HB (DD HCC)	Carbo, Etop, Cis, Dox, Ifo	(1) Liver (R1)	19
2 (D)	36	III M+	HB	Carbo, Etop, Melp	(1) Liver (R0), (2) Lung (MD), (3) Lung (R0)	33
3 (A)	16	III M+F+	HB	Carbo, Etop	(1) Liver (R0) (+Lung—NAD)	22
4 (D)	4	IV M+E+	HB	Cis, Dox, Ifo, Carbo, Etop, Ir	(1) Liver (R0), (2) Lung (MD)	7
5 (A)	26	III M+	HB	Cis, Do, Ifo, Carbo, Etop	(1) Liver (R0), (2) Lung R + L (R0)	19
6 (D)	13	IV M+	HB	Cis, Dox, Ifo, Carbo, Etop	(1) Liver (R0)	8
7 (D)	3	III R+	SCUD (INI1 MD)	Cis, Dox, Ifo	(1) Liver (R1)	9
8 (A)	10	IV R+F+	HB	Carbo, Etop	(1) Liver (R0)	7
9 (A)	12	II	HB	Cis, Dox, Ifo	(1) Liver (R0)	21
10 (D)	117	II R+	HB (DD HCC)	Cis, Dox, Ifo	(1) Liver (R0)	45
11 (D)	53	IV M+P+	HB (DD HCC)	Carbo, Etop	(1) Liver (R1), (2) Lung (R0), (3) Lung (R0)	12
12 (A)	158	II P+	HB (Macrotrab)	Cis, Dox, Ifo	(1) Liver (R0)	21
13 (D)	40	III	HB	Cis	(1) Liver (R0)	5
14 (A)	18	I M+E+	HB	Cis, Carbo, Dox, Etop, Ifo, Vin, Ir	(1) Liver (R0), (2) Lung (R1), (3) Lung (R0)	66
15 (A)	23	III M+P+	HB	Cis, Carbo, Dox	(1) Liver (R1)	13
16 (D)	24	II	HB	Cis, 5-FU, Vin	(1) Liver (R0)	10
17 (A)	94	IV M+F+	HB	Cis, Carbo, Dox	(1) Liver (R0)	21
18 (D)	35	IV P+F+	HB	Cis, Carbo, Dox	(1) LTx (R1)	10
19 (D)	21	III V+P+F+	HB	Cis, Carbo, Dox	(1) Liver (R0), (2) Lung (R0)	8
20 (A)	8	II	HB	Cis	(1) Liver (R0)	10
21 (A)	15	III	HB	Cis, Carbo, Dox, Vin, Ir	(1) Liver (R0), (2) Lung (R0), (3) Lung (R0)	10
22 (A)	29	IV M+V+P+F+	HB	Cis, Carbo, Dox, Vin, Ir, Ifo, Etop	(1) Lung R (MD) and L (R0), (2) LTx (R2)	35
23 (A)	75	IV V+F+	HB (DD HCC)	Cis, Carbo, Dox, Sor, Vin, Ir	(1) Liver (R1), (2) Lung (R1)	12
24 (A)	51	IV F+	HB/SCUD (INI1+)	Cis, Carbo, Dox	(1) Liver (R0)	26
25 (D)	4	III V+	HB	Cis, Carbo, Dox, Vin, Ir, Etop	(1) Liver (R2), (2) LTx (R0)	13

A = alive, D = dead, MD = missing data, NAD = no abnormality detected, Pat. = patient, PRETEXT = pre-treatment extent of disease, M+ = distant metastatic disease, V+ = venous invasion, P+ = portal vein invasion, E+ = extrahepatic intra-abdominal mets, R+ = tumor rupture, F+ = multi-focal, N+ = lymph node mets, Cis = cisplatin, Carbo = carboplatin, Etop = etoposide, Dox = doxorubicin, Melp = melphalan, Ifo = ifosfomide, Vin = vincristine, Ir = irinotecan, 5-FU = 5-fluorouracil, Sor = sorafenib, R0 = microscopically complete resection, R1 = microscopically positive resection, R2 = macroscopically positive resection, Dx = diagnosis, m = months, LTx = liver transplant.

**Table 3 cancers-16-00696-t003:** Relapse Location, Relapse Treatment, Event-Free Survival, Overall Survival, and Status at Last Follow-Up for the 25 Relapse Patients.

Pat. No.	Relapse Location	Relapse Chemo	Surgery for Relapse	EFS/OS from Relapse (m)	Status
1	Local	Dox, To	Liver (R0)	13/40	D—FR
2	Met-P	Tro, Id (Pall)	Peritoneum (R2)	3/10	D-Prog
3	Met-L	Cis, Dox, Ifo	Lung (R0)	102/102	2nd CR
4	Met-P	No Chemo	Peritoneum (R0)	0/2	D-Prog
5	Met-L	Cis, Melp, Ir, Id, Bu	Lung (R0)	171/171	2nd CR
6	Met-P	Cis, Dox, Ifo (Pall)	Peritoneum (R2)	2/2	D-Prog
7	Met-P	Carbo, Etop	Peritoneum (R1)	18/29	D—FR
8	Local	Cis, Carbo, Dox, Ifo	Liver (R0)	186/186	2nd CR
9	Met-L	Cis, Carbo, Dox, Ifo, Etop	Lung (R0)	76/76	2nd CR
10	Local + P	Carbo, Ifo, Etop, Beva, Sor, Sun	Liver + Peritoneum (MD), Liver (MD)	8/37	D—Prog
11	Met-L	Carbo, Dox, Ifo	No Op	80/146	D—FR
12	Local	Carbo, Etop, Ir, Beva	Liver (R0), LTx (R0)	64/64	2nd CR
13	Local + P + L	Carbo, Dox, Ifo, Etop, 5FU, Vin, Im	No Op (R3)	2/8	D—Prog
14	Met-L	Carbo, Ifo, Etop, Ir, Vin	Lung (R0)	15/15	2nd CR
15	Met-L	Ir, Vin	Lung (R0)	60/60	2nd CR
16	Met-CNS	No Chemo	No Op	0/0	D
17	Local	Ir, Vin	LTx (R0)	55/55	2nd CR
18	Local	No Chemo	LTx (R1)	1/1	D
19	Met-L	Cis, Carbo, Dox, Vin, Ir	Lung (R1), Lung (NAD), Liver (R0)	4/16	D—Prog
20	Met-L	Carbo, Dox	Lung (R0)	66/66	2nd CR
21	Met-L	Cis, Dox, 5FU, Vin	Lung (R0)	30/30	2nd CR
22	Met-L	Vin (oral), Ir	Lung (R0), Lung (MD)	14/39	5th CR
23	Met-CNS	Carbo, Etop, Melp, Ir, Vin	CNS (R1)	12/16	2nd Rel
24	Local	Carbo, Etop, Vin, Ir	Liver (R0)	81/81	2nd CR
25	Met-L	Carbo, Etop	Lung (R0)	2/10	D—Prog

P = peritoneal, L = lung, Met = metastatic disease, MD = missing data, Cis = cisplatin, To = topetecan, Tro = trofosfamid, Id = idarubicin, Pall = palliative, Carbo = carboplatin, Etop = etoposide, Dox = doxorubicin, Melp = melphalan, Bu = busulphan, Ifo = ifosfomide, Beva = bevacizumab, Sor = sorafenib, Sun = sunitinib, Vin = vincristine, Ir = irinotecan, 5-FU = 5-fluorouracil, Im = imatinib, R0 = microscopically complete resection, R1 = microscopically positive resection, R2 = macroscopically positive resection, R3 = biopsy only, Dx = diagnosis, m = months, D = death, FR = further relapse, Prog = progression, CR = complete remission, Rel = relapse.

## Data Availability

The data presented in this study are available on request from the corresponding author. The data are not publicly available as restrictions apply to the availability of these data.

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
