# Peer review of "Hepatoblastoma Relapse—Findings from the German HB99 Trial and the German Liver Tumor Registry"

_cancers, 2024, doi:10.3390/cancers16040696_

Round 1

Reviewer 1 Report

Comments and Suggestions for Authors

The title of the manuscript "Hepatoblastoma Relapse Treatment Recommendations based on findings from the German HB99 trial and the German Liver Tumour Register" is misleading since no recommendations are finally proposed. It should be changed or reworded to "Hepatoblastoma Relapse - findings from the German HB99 trial and the German Liver Tumour Register"

Material and methods                                                                                        line 92 - please clarify the time of tumor resection - during? or postchemotherapy ,what does it mean during.                                                  line 98 - there is a difference whether a patient has a R1 resection or macroscopic residual, were there relapsed patients who had makroscopical residual disease if yes what was the histology ?                                                   The text in lines 105-112 should be revised starting with a definition of a relapse. None of the events in this paragraph are clearly characterized. In line 113 the author defines progression - does it concern progression of relapse?? This needs revision. Moreover the authors do not have to state that patients who had progressive disease were excluded - from the definition of relapse it is clear that they had achieved CR.

Results.                                                                                                           Table 1 is acceptable and allows to compare the data from other studies but another table/tables  containing information on each of the 25 patients ( age, stage, type of chemotherapy, surgery for primary disease, time from diagnosis to relapse, type of relapse, type of chemotherapy and surgery for relapse, follow-up from the time of relapse - alive without disease/with disease, death) are adviced. Such  table/tables would be informative and allow  to follow  the results with comprehension. The results are badly presented .  The lenghty descriptions with a lot of information ( more or less important ) do not help to understand how these patients were treated of their primary disease and relapse, who received what kind of primary or relapse treatment. The  role of irinotecan is far- fetched ( 9 pts received it, 11 didn't- no statistical difference in OS) though in practice it is used in a relapse setting. All in all the results should be rewritten. The results more or less confirm the findings from other groups. 

The discussion also needs thorough revision (should follow the results). It doesn't have to be so robust otherwise the reader gets lost in details. 

To sum up

- the results from the study do not allow to formulate recommendations for the treatment of HBL relapse

- eventhough the topic of this study is not new there is still room for such manuscripts provided that it is rewritten and with the revision of the title. 

Comments on the Quality of English Language

Editing of English language is adviced.

Author Response

Thank you for your feedback on the article about hepatoblastoma relapse. We would like to resubmit an amended report on hepatoblastoma:

With regards to the comments from reviewer 1:

  • Extensive editing of the English language required – This paper was written by a university educated native English speaker (Rebecca Maxwell). I have reviewed the article and have attempted to make the wording clearer and more concise.
  • As suggested, the title of the manuscript has been changed to reflect the fact that no specific recommendations have been proposed. The title is now “Hepatoblastoma Relapse – Findings from the German HB99 trial and the German Liver Tumour Registry”.
  • Line 92 –The wording “during chemotherapy” is used here to reflect the fact that the resection does not always take place at the end of a multi-cycle chemotherapy protocol. The resection can also occur after any of the individual chemotherapy cycles that make up the treatment protocol and thus before the end of the chemotherapy treatment. I have amended this sentence so that it now reads “Tumours were usually resected following one of the planned chemotherapy cycles or after all of the chemotherapy cycles from the chosen protocol had been given”.
  • Line 98 – Were there patients who had macroscopic residual disease? If yes, what was the histology? Prior to relapse there were 2 patients who were graded as having macroscopic residual disease after the operation. The first of these patients had a liver transplant where potential abdominal seeding was noted during the operation. The liver histology here was reported as a mixed epithelial (foetal)-mesenchymal HB. The other patient was graded as having macroscopic residual disease post-op following a liver resection. This patient had been noted to have multiple liver lesions at diagnosis but it was felt at that time that only one of these lesions was cancerous. At the time of the initial resection, 3 lesions were removed from the liver, the suspicious lesion was confirmed to be an epithelial (principally foetal) HB. The other lesions were identified as a capillary haemangioma and a pseudocyst. After the operation the AFP initially dropped and then started to increase again and so it was concluded that some of the remaining lesions must also be malignant. A subsequent liver transplant was performed and the AFP subsequently normalised. The definition of R1 and R2 resections has been added to the paper and now reads “R0 resections are those where there was a microscopically complete resection, R1 resections were microscopically incomplete resections and R2 resections were macroscopically incomplete resections”.
  • Lines 105-112 – The definitions of relapse, complete remission, overall survival and event free survival have been amended as requested to make them more precise.
  • Line 113 – The definition of progression has been amended to reflect the fact that it means progression post-relapse.
  • The sentence with regards to the exclusion of patients who had progression after their initial treatment and never achieved a complete remission has been removed.
  • New results tables have been added as requested so as to make it easier to follow the results.
  • The results section has been reviewed and edited in order to highlight the more important results.
  • Thank you for your comments with regards to irinotecan. We feel that although there was no statistically significant difference in survival between those who received irinotecan in relapse and those who did not, that this is still an important area of discussion within the hepatoblastoma field. We have changed the wording used in the paper to reflect the fact that further research is needed here before a recommendation can be given (line 315).
  • The discussion section has been revised so that is contains the most relevant information and is easier to follow.

Reviewer 2 Report

Comments and Suggestions for Authors

There were some technical problems with submission of my first review so here are the key comments:

This is a well written paper with very impressive detailed data on all the relapse Hepatoblastoma in Germany. So, a good job is done. Data on relapse HB is indeed sparse so highly needed and relevant to present. Hopefully the data are added to the international ongoing RELIVE database?

My major concern is that nearly all extracted data are provided and you risk that most readers will not reach the end of this manuscript. There is simply too much information and a more focused manuscript is needed. Especially, the results section and discussion, I feel, should be shortened significantly. I believe that some prioritization among all the data would clearly benefit this relevant paper.  Eg, add a suppl .appendix with the result you decide omit in the result section. At this stage it does not make sense to go into details for this paper until this is done.

Our comments

The title: ‘Hepatoblastoma Relapse Treatment Recommendations based on findings from the German HB99 trial and the German Liver Tumour Register’ seems suboptimal as the paper does not provide clear cut treatment guidelines.  I would change it to eg, ‘Characteristics and outcome (Or Epidemiology?) of Hepatoblastoma Relapse - the German; HB99 trial and Liver Tumour Register’

Abstract: Delete the first two very long sentences and focus on some more results.

In conclusion, suggest to prioritize and delete a significant amount of data/ result and keep the discussion concise. Then the paper will become more interesting and read much better . 

Comments on the Quality of English Language

Good

Author Response

With regards to the comments from reviewer 2:

  • Minor editing of the English language required – This paper was written by a university educated native English speaker (Rebecca Maxwell). I have reviewed the article and have attempted to make the wording clearer and more concise.
  • This data has been added to the RELIVE database.
  • The results and discussion sections of the paper have been reviewed and amended so as to be provided a more focused overview of the most important findings.
  • The title of the paper has been changed to reflect that fact that the paper does not provide clear cut treatment guidelines. The title is now “Hepatoblastoma Relapse – Findings from the German HB99 trial and the German Liver Tumour Registry”.

Round 2

Reviewer 1 Report

Comments and Suggestions for Authors

My recommendations are to accept the manuscript in the present form.